# Regional sequencing collaboration reveals persistence of the T12 *Vibrio cholerae* O1 lineage in West Africa

Eme Ekeng[1†], Serges Tchatchouang[2†], Blaise Akenji[3], Bassira Boubacar Issaka[4], Ifeoluwa Akintayo[5], Christopher Chukwu[1], Ibrahim Dan Dano[4], Sylvie Melingui[3], Sani Ousmane[4], Michael Oladotun Popoola[1], Ariane Nzouankeu[2], Yap Boum[6], Francisco Luquero[6], Anthony Ahumibe[1], Dhamari Naidoo[7], Andrew Azman[8], Justin Lessler[8], Shirlee Wohl[8]*

[1]Nigeria Centre for Disease Control, Abuja, Nigeria; [2]Centre Pasteur du Cameroun, Yaoundé, Cameroon; [3]National Public Health Laboratory, Yaoundé, Cameroon; [4]Centre de Recherche Médicale et Sanitaire, Niamey, Niger; [5]Department of Pharmaceutical Microbiology, Faculty of Pharmacy, University of Ibadan, Ibadan, Nigeria; [6]Epicentre, Paris, France; [7]World Health Organization Nigeria, Abuja, Nigeria; [8]Department of Epidemiology, Johns Hopkins Bloomberg School of Public Health, Baltimore, United States

*For correspondence:
swohl@jhu.edu

†These authors contributed equally to this work

Competing interests: The authors declare that no competing interests exist.

## Abstract

**Background:** Despite recent insights into cholera transmission patterns in Africa, regional and local dynamics in West Africa—where cholera outbreaks occur every few years—are still poorly understood. Coordinated genomic surveillance of *Vibrio cholerae* in the areas most affected may reveal transmission patterns important for cholera control.

**Methods:** During a regional sequencing workshop in Nigeria, we sequenced 46 recent *V. cholerae* isolates from Cameroon, Niger, and Nigeria (37 from 2018 to 2019) to better understand the relationship between the *V. cholerae* bacterium circulating in these three countries.

**Results:** From these isolates, we generated 44 whole *Vibrio cholerae* O1 sequences and analyzed them in the context of 1280 published *V. cholerae* O1 genomes. All sequences belonged to the T12 *V. cholerae* seventh pandemic lineage.

**Conclusions:** Phylogenetic analysis of newly generated and previously published *V. cholerae* genomes suggested that the T12 lineage has been continuously transmitted within West Africa since it was first observed in the region in 2009, despite lack of reported cholera in the intervening years. The results from this regional sequencing effort provide a model for future regionally coordinated surveillance efforts.

**Funding:** Funding for this project was provided by Bill and Melinda Gates Foundation OPP1195157.

## Introduction

Molecular characterization of seventh pandemic *Vibrio cholerae* has led to new insights about global cholera transmission and has highlighted the important role of transmission within and between Asia and Africa (*Weill et al., 2017*). Combining genomic and epidemiological data may allow us to better understand the dynamics of ongoing outbreaks, and advances in sequencing technology have recently made whole genome sequencing more feasible even in low-resource settings.

Although recent studies have used whole genome sequences of *V. cholerae* O1 to better understand the global movement patterns of seventh pandemic cholera in sub-Saharan Africa and

elsewhere (*Domman et al., 2017*; *Mutreja et al., 2011*; *Weill et al., 2019*; *Wick et al., 2017*), regional and local dynamics in West Africa—which reports cholera regularly—are still poorly understood. Epidemiological data suggest outbreaks across this region may be connected (*UNICEF, 2013a*; *UNICEF, 2013b*; *UNICEF, 2013c*), but the nature of recurring outbreaks in the region has yet to be understood. It is unclear if cases go unreported in years between outbreaks or if each represents a distinct pandemic *V. cholerae* introduction from outside the region. Discerning between these two transmission scenarios is important for the development of locally adapted and effective cholera control and prevention strategies.

One way to distinguish between transmission scenarios is to compare the specific *V. cholerae* lineages in circulation. Previous studies initially identified three waves of seventh pandemic *V. cholerae* (*Mutreja et al., 2011*), which were later broken down into 13 plausible introduction events from Asia into Africa (*Weill et al., 2019*; *Wick et al., 2017*), termed T1–T13. These studies used sequencing of historical isolates to identify the predominant lineages circulating in different regions of the African continent since 1970, providing important information about global and continent-wide circulation patterns. In this manuscript, we aim to use additional sequence data from recent cholera outbreaks to improve our understanding of more detailed *V. cholerae* transmission patterns within West Africa, and to understand if regionally coordinated surveillance and response may improve containment outcomes.

## Approach: a regional sequencing collaboration

In October 2019, researchers from Cameroon, Niger, Nigeria, and the United States came together to discuss how whole genome sequencing could contribute to understanding of cholera spread in these countries. Participants brought *V. cholerae* isolates (collected from medically attended cholera cases in cholera treatment centers or health centers in their countries) to the one-week workshop in Abuja, Nigeria, and were trained in pathogen sequencing methods using the portable Oxford Nanopore MinION platform. The workshop cohort was predominantly laboratory scientists from regional and national health laboratories who specialize in *V. cholerae* and microbiology techniques; individuals who were well-poised to master the laboratory demands of whole genome sequencing and had a deep understanding of cholera in the region. Bioinformatic analysis was also a central component of the training, and participants focused on analyzing data generated during the workshop on specialized laptop computers. Sequencing and analysis of recent isolates during the training allowed for joint, regionally focused interpretation of the data during and after the workshop.

In addition to the practical, hands-on sequencing training, a key component of this workshop was bringing together personal experience and detailed epidemiological data from the three countries, which we used to select the specific *V. cholerae* isolates to be sequenced during the workshop — isolates that would provide the best understanding of the 2018–2019 outbreak. During this outbreak, Cameroon, Niger, and Nigeria all reported large numbers of cholera cases (2066, 3083, and 32,752 cases, respectively) (*Nigeria Centre for Disease Control, 2019*; *Regional Cholera Platform in West and Central Africa, 2012*) despite several years of low reported cholera incidence in the region (*Figure 1A*). In all countries, reported cases were defined according to the World Health Organization suspected case definition (*Global Task Force on Cholera Control Surveillance Working Group, 2017*): individuals with acute watery diarrhoea and severe dehydration in areas where a cholera outbreak has not been declared, and individuals with acute watery diarrhoea in areas where an outbreak has been declared.

To better understand the history of cholera transmission in the region, we spent the beginning of the workshop discussing previous sequencing efforts of historical *V. cholerae* isolates from the region and existing gaps in our understanding of transmission. Previously published sequences from the three countries include 45 sequences from Cameroon from 1970 to 2011, 15 sequences from Niger from 1970 to 2010, and 22 sequences from Nigeria from 1970 to 2014 (*Weill et al., 2019*; *Wick et al., 2017*). These data show that the T1 lineage was the predominant lineage circulating in Cameroon, Nigeria, and Nigeria in 1970, followed by the T7 and T9 lineages in the 1990s and early 2000s (*Weill et al., 2019*; *Wick et al., 2017*). Since 2009, only T12 has been observed in these countries, though at the start of the workshop prior sequences were available only through 2014. In any given year, the circulating *V. cholerae* lineage was the same in Cameroon, Niger, and Nigeria (*Weill et al., 2019*; *Wick et al., 2017*), providing additional evidence that outbreaks in these countries are connected. As five years had elapsed since the last published *V. cholerae* O1 genome in the

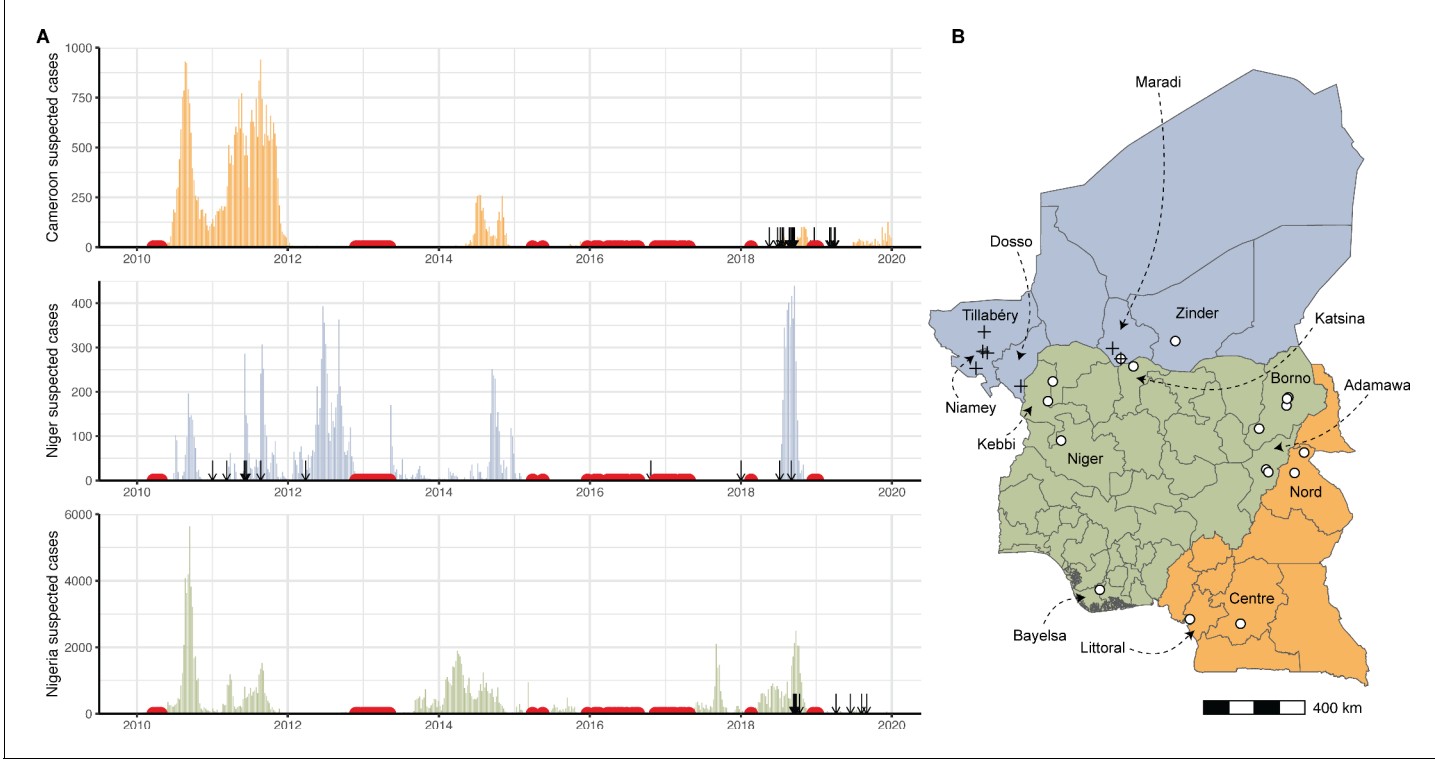

**Figure 1.** Cholera cases and sequenced isolates. (**A**) Weekly suspected cholera cases for Cameroon (orange), Niger (blue), and Nigeria (green) from 2010 through 2019 ('Regional Cholera Platform in West and Central Africa,' n.d.) (***Figure 1—source data 1***). Suspected cases are defined as recommended by the World Health Organization (***Global Task Force on Cholera Control Surveillance Working Group, 2017***). Red points: weeks with no more than five suspected cases reported across all three countries. Arrows: collection dates of isolates sequenced. Collection dates provided as year only (n=2) are plotted on January 1 of their given year. (**B**) Map of Cameroon, Niger, and Nigeria. Colors are as in (**A**). White points: location of sequenced isolates collected in 2018 and 2019 (***Figure 1—source data 2***). Black crosses: location of sequenced isolates collected prior to 2018. One isolate with unknown sub-country location is not shown.

The online version of this article includes the following source data for figure 1:

**Source data 1.** Case counts by epidemiological week in Niger, Nigeria, and Cameroon, 2010 to 2019.
**Source data 2.** Sample metadata for isolates sequenced in this study.

region, we aimed to generate a contemporary snapshot of recent *V. cholerae* O1 outbreaks to determine if recent cases were due to a new introduction of the pathogen, and to gain new insights into regional cholera transmission dynamics and spread of antibiotic resistance. Simultaneously, we discussed plans for building a regional cholera genomics network to help inform cholera preparedness and outbreak response in the future.

## Materials and methods

### Sample collection and cholera confirmation

*V. cholerae* isolates were collected from clinically confirmed cholera cases in Cameroon, Niger, and Nigeria between 2011 and 2019 (***Figure 1—source data 2***). In Cameroon, isolates came from cholera treatment centers from throughout the country. In Niger, isolates came from health centers and cholera treatment centers during outbreaks in the three southern provinces (Dosso, Maradi and Zinder), as well as from the province of Tillabery. And in Nigeria, isolates came from cholera treatment centers in the Northeastern and Northwestern regions of the country. In all three countries, suspected cholera stools were cultured on TCBS medium. Phenotypic identification of *V. cholerae* colonies was based on morphology, motility, and biochemical characteristics (positive oxidase, saccharose, indole, and gelatinase). *V. cholerae* isolates were confirmed with agglutination tests with

anti-O1 or anti-O139 serum (WHO antisera and Denka Seiken Agglutinating Sera *Vibrio cholerae* Antisera Set). Two isolates from Nigeria that did not show agglutination were identified as non-O1 *Vibrio cholerae* through Gram staining (Gram negative), oxidase testing (positive), and the API (Analytical Profile Index) 20E kit at the National Reference Laboratory in Nigeria.

## DNA extraction and quantification

DNA extraction was performed at Centre Pasteur du Cameroun (Yaounde, Cameroun), Centre de Recherche Médicale et Sanitaire (Niamey, Niger), and at National Reference Laboratory (NRL; Nigeria Centre for Disease Control, Abuja, Nigeria). Confirmed *V. cholerae* isolates were cultured on selective TCBS, MH and/or HCK agar plates and incubated at 37°C overnight. In all countries, *V. cholerae* DNA was extracted following the standard protocol from the Qiagen QIAamp DNA Mini Kit with a final elution volume of 200 µL. Extracted DNA from *V. cholerae* isolates were stored in 1.5 mL Eppendorf tubes at 4°C or at room temperature until quantification during the workshop at NRL. DNA concentrations were measured with the Qubit Fluorometer 4.0 (Thermo Fisher) using the dsDNA HS assay standard protocol. Input concentrations for each isolate are recorded in *Figure 2— source data 1*.

## Oxford nanopore library construction and sequencing

Library preparation and sequencing was performed at the NRL or at Johns Hopkins University (Baltimore, Maryland, United States). In both locations, quantified, extracted DNA was diluted to 1000 ng in 48 µL input material, as required by the SQK-LSK109 library preparation kit from Oxford Nanopore Technologies. For samples with a concentration below 20 ng/µL, 48 µL of extracted DNA was used, regardless of the final amount (<1000 ng).

DNA repair and end-prep was performed with 2 µL NEBNext FFPE DNA Repair Mix, 3.5 µL NEBNext FFPE DNA Repair Buffer, 3 µL Ultra II End-Prep Reaction Mix, and 3.5 µL Ultra II End-Prep Enzyme Buffer (New England Biolabs). The reaction was incubated at 20°C for 20 min and then 65°C for 10 min, and then purified with 1x AMPure XP beads (Beckman Coulter) and eluted in 25 µL Elution Buffer (Promega). A unique 2.5 µL Native Barcode (Oxford Nanopore EXP-NBD104) and 25 µL Blunt/TA Ligase Master Mix (New England Biolabs) were added to dA-tailed DNA and the reaction mixture was incubated for 30 min at room temperature. As before, the reaction was purified with 1x AMPure XP beads. The purified DNA was eluted in 10 µL Elution Buffer. Equimolar amounts of six to nine isolates were pooled to an approximate total DNA amount of 1000 ng, and the pool was diluted to 65 µL in Elution Buffer. The barcoded pool was added to 5 µL Adapter Mix II (Oxford Nanopore EXP-NBD104), 20 µL NEBNext Quick Ligation Reaction Buffer (New England Biolabs), and 10 µL Quick T4 DNA Ligase (New England Biolabs). The reaction was incubated at room temperature for 30 min and purified with 0.5x AMPure XP beads, using Long Fragment Buffer (Oxford Nanopore SQK-LSK109) instead of ethanol in the wash steps. The final library was eluted in 15 µL Elution Buffer and quantified using the Qubit Fluorometer 4.0 to confirm the process had been successful.

For sequencing, 190 ng pooled final library was diluted to 12 µL in Elution Buffer and combined with 37.5 µL Sequencing Buffer and 25.5 µL Loading Beads (Oxford Nanopore SQK-LSK109) before loading onto a primed R9.4.1 Oxford Nanopore MinION flowcell. For each sequencing run, the MinION was run for 48 hr using an offline-capable version of MinKNOW made available by Oxford Nanopore Technologies specifically for this project.

All sequencing files were basecalled using Guppy version 3.0.3 with the flip-flop model (dna_r9.4.1_450bps_fast.cfg). Adapters were removed from reads and reads were demultiplexed into individual samples with Porechop (*Wick et al., 2017*). Filtlong (https://github.com/rrwick/Filtlong) was used with the following options to reduce file size and filter out low quality sequencing reads: '–keep_percent 90 –target_bases 800000000.' The resulting filtered FASTQ files were used as input for all subsequent analyses, including assembly.

## Reference-based genome assembly

Reference-based assembly was performed with nanopolish version 0.11.1 (*Loman et al., 2015*) using the seventh pandemic O1 strain N16961 as a reference (accession: AE003852/AE003853). Briefly, reads were indexed with nanopolish, then mapped to the reference with minimap2 version 2.17 (*Li, 2018*), then sorted and indexed with samtools version 1.10 (*Li et al., 2009*). Variants to the

reference genome were then called with nanopolish, and bcftools version 1.9 (*Li, 2011*) was used to filter out variants with less than 75% support and sites with less than 100x coverage. A custom Python3 script was used to create consensus genomes from the VCF files, keeping only sites with at least 100x coverage. The full reference-based genome assembly pipeline for *V. cholerae* is publicly available at: https://github.com/HopkinsIDD/minion-vc (copy archived at swh:1:dir: 3180fe441b99128a760fac987cf02a5b4e9bbdaf).

## Maximum likelihood estimation

A maximum likelihood tree was generated on a multiple sequence alignment of the genomes from Cameroon, Niger, and Nigeria as well as 1280 previously published *V. cholerae* whole genome sequences (*Figure 2—source data 3*; *Bwire et al., 2018*; *Weill et al., 2019*; *Wick et al., 2017*). Recombinant sites were masked as previously described (*Weill et al., 2017*) (see https://figshare. com/s/d6c1c6f02eac0c9c871e for masking sites) and using Gubbins version 2.3.4 (*Croucher et al., 2015*). The maximum likelihood tree was generated on the resulting masked SNP alignment using IQ-TREE version 1.6.10 (*Nguyen et al., 2015*) with a GTR substitution model and 1000 bootstrap iterations. The tree was rooted on A6 (accession: ERR025382). Trees were visualized in FigTree version 1.4.4 (http://tree.bio.ed.ac.uk/software/figtree/).

## *Vibrio cholerae* typing

Reads from each isolate were mapped to reference sequences *ctxA* (accession: AF463401), *wbeO1* (accession: KC152957), and *wbfO139* (accession: AB012956), as described in *Greig et al., 2018*. Reads were mapped using minimap2 version 2.17 (*Li, 2018*) and then sorted with samtools version 1.10 (*Li et al., 2009*). The number of reads mapping at each reference gene site was counted using samtools.

## Antibiotic susceptibility assays

Antibiotic susceptibility testing of *V. cholerae* isolates were done by disc diffusion on MH agar plates at Centre Pasteur du Cameroun, Centre de Recherche Médicale et Sanitaire, and NRL. The following antibiotics were tested for resistance: amoxicillin/clavulanic acid, nalidixic acid, polymyxin, tetracycline, ciprofloxacin, and chloramphenicol. The interpretation of resistance/susceptibility profiles was done according to the CASFM 2007 guidelines for Enterobacteriaceae.

## Antibiotic resistance gene detection

Genes associated with antibiotic resistance were detected in two ways: first, by using the software abricate (https://github.com/tseemann/abricate (copy archived at swh:1:dir: 822d8a8864813a025f248c85d422f5b6a82a5b2a)) with the NCBI (*Feldgarden et al., 2019*) and CARD (*Jia et al., 2017*) databases. And second, by translating the nucleotide sequences on the EMBOSS (*Rice et al., 2000*) website and aligning the resulting translations to target genes downloaded from https://pubmlst.org/vcholerae/ and https://www.ncbi.nlm.nih.gov/ with the CLUSTALW (*Thompson et al., 1994*) website. Resistance to quinolones was determined by comparing the assemblies to reference strains, as described in previous studies (*Chien et al., 2016*; *Divya et al., 2014*; *Zhou et al., 2013*).

## Results

During isolate selection, we prioritized isolates from locations near the borders between Cameroon, Niger, and Nigeria, in order to increase the likelihood of capturing evidence of cross-border transmission. We selected 46 isolates from clinical and lab confirmed cases for sequencing (16 from Cameroon, 15 from Niger, 15 from Nigeria), 37 of which were from 2018 to 2019 cases (*Figure 1B*, *Figure 1—source data 2*).

During the workshop, we sequenced these isolates on the Oxford Nanopore MinION platform, producing 44 genomes covering at least 95% of the N16961 reference at >100x coverage (*Figure 2*, *Figure 2—source data 1*). We aligned these genomes to 1280 previously published *V. cholerae* O1 whole genome sequences (*Bwire et al., 2018*; *Irenge et al., 2020*; *Weill et al., 2019*; *Wick et al., 2017*), which provide a broad overview of global seventh pandemic O1 cholera diversity. The

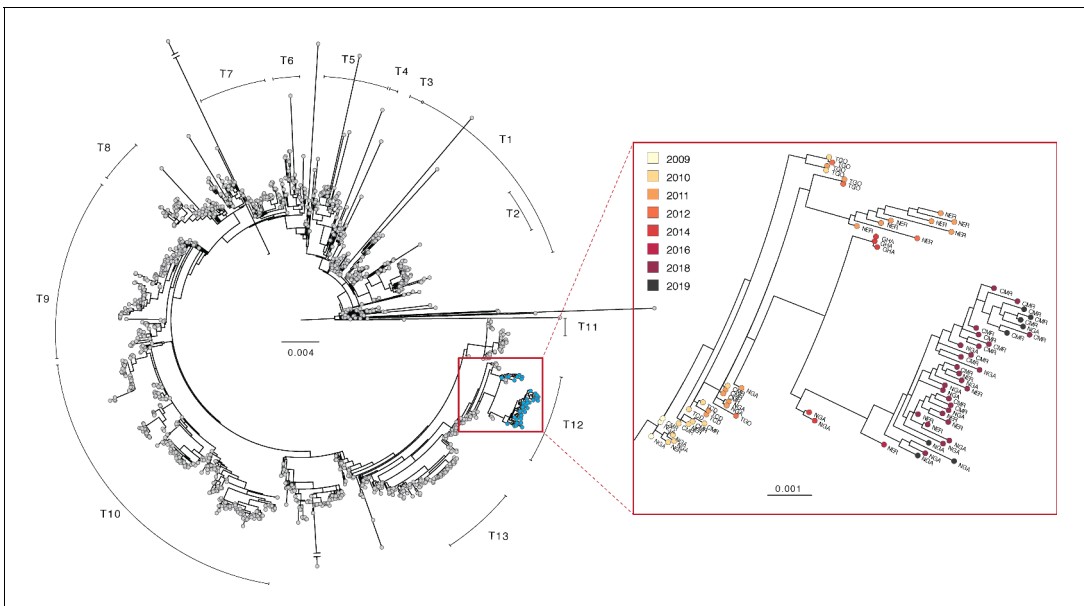

**Figure 2.** Phylogenetic tree of *V. cholerae* O1 sequences. Left: maximum likelihood tree of global *V. cholerae* isolates. Samples generated in this study are shown in blue (see also, *Figure 2—source data 1*). Right: zoom view of a portion of the T12 lineage containing *V. cholerae* genomes generated in this study. Country codes: TGO, Togo; NER, Niger; GHA, Ghana; CMR, Cameroon; NGA, Nigeria; TCD, Chad. Scale bar unit: nucleotide substitutions per site.

The online version of this article includes the following source data and figure supplement(s) for figure 2:

**Source data 1.** Table of sequencing metrics for *V. cholerae* O1 genomes.

**Source data 2.** Summary table of published *V. cholerae* O1 genomes included in this study.

**Source data 3.** Accession numbers, references, and basic metadata for sequences included in global phylogeny.

**Source data 4.** Cases reported to the World Health Organization by decade, continent, and African subregion (where applicable), compiled from weekly epidemiological records (*World Health Organization, 2019*).

**Figure supplement 1.** Proportion of reported cholera cases and *V. cholerae* sequences by continent.

**Figure supplement 2.** Proportion of reported cholera cases and *V. cholerae* sequences by African region.

majority of these sequences are from Africa and South Asia, and reported cases in West and Central Africa (*World Health Organization, 2019*) are well represented by the sequence data (*Figure 2—source data 2*, *Figure 2—figure supplement 1*, *Figure 2—figure supplement 2*). We generated a maximum likelihood tree from these data, which showed that all 44 isolates belong to the T12 lineage. Two sequenced isolates did not align to the reference (NGA_148_2019, NGA_252_2019), and we found that the reads did not map to the *wbe* or *wbf* gene clusters associated with the O1 and O139 serogroups, respectively (or the *ctxA* cholera toxin) (*Greig et al., 2018*). These results are concordant with laboratory testing performed in Nigeria, which designated these isolates as non-O1 *V. cholerae*. We also performed antimicrobial resistance (AMR) gene detection on these sequences and compared the results to previously published sequences. AMR profiles were similar to prior sequences from this region, though we detected new mutations in genes (*gyrA* and *gyrB*) associated with quinolone resistance (see *Supplementary file 1* for AMR results).

## Discussion

We found no evidence for a new introduction of *V. cholerae* O1 from outside of West Africa between 2014—the date of the last previously published isolate from these countries—and 2018. This finding is based on the fact that all newly sequenced isolates from 2018 to 2019 (as well as the 2011–2016 isolates from Niger) fall within the T12 lineage, which was present in West (and parts of Central) Africa from 2009 to 2014. Instead, we found evidence for evolution of the bacterium within the region, as the 2018–2019 sequences are part of a sub-clade containing 2014 and 2016 isolates from countries in West Africa (*Figure 2*). Within this sub-clade, the 2018–2019 sequences are

distinct. In concordance with the epidemiologic data, the high similarity between sequences (including shared mutations in known AMR genes, see *Supplementary file 1A and 1B*) from the three countries suggests that their outbreaks are linked.

The phylogeny also suggests that nearby countries such as Ghana and Togo may have outbreaks connected to those in Cameroon, Niger, and Nigeria, although it is unclear just how far this cholera transmission region extends. As of this writing, there are no published *V. cholerae* O1 genomes from the nearby countries of Benin, Burkina Faso, or Mali, for example, from after 2010. Sequences from Central African Republic, South Sudan and Democratic Republic of the Congo from 2011 to 2018 suggest that the T10 lineage is predominant in Central Africa (*Irenge et al., 2020*; *Weill et al., 2019*), but more recent data are needed to delineate the boundary between T10 and T12 circulation.

Even so, there is strong evidence that, at least in Cameroon, Niger, and Nigeria, T12 *V. cholerae* was present throughout 2009–2019 despite few reported cases in 2014–2017. This could be explained by underreporting or under-diagnosing of cases, movement of the same cholera lineage in and out of these countries during that time period, or an environmental reservoir of *V. cholerae* in the region (such as Lake Chad, although prior studies have provided evidence against this *Debes et al., 2016*) that is responsible for repeated reintroduction of the same lineage back into the population.

Expanded surveillance and additional sequencing data will be necessary to understand the interconnectedness of populations across countries and achieve global elimination goals such as those outlined in the Global Task Force for Cholera Control Roadmap (*Global Task Force for Cholera Control, 2017* ). For example, limiting regional movement could be an ineffective strategy if new lineages are repeatedly and simultaneously introduced from outside the region. Additionally, sequencing data can be used to detect and track resistance to antibiotic treatments, and to identify potential new resistance-conferring mutations present in local isolates (*Mashe et al., 2020*). A regional cholera surveillance network could facilitate efforts to gather isolates, build the local capacity needed to generate sequencing data, and ensure that these data are interpreted within the regional context.

As a result of the workshop, scientists from Cameroon, Niger, and Nigeria have all been trained in *V. cholerae* sequencing with the Oxford Nanopore MinION platform, and scientists at Nigeria Centre for Disease Control National Reference Laboratory have used this expertise to continue sequencing *V. cholerae* and other pathogens. Challenges still exist to continued development of sequencing capacity in West Africa, including the need for additional bioinformatics training and the lack of established supply chains for sequencing reagents. Through subsequent training workshops and ongoing collaborations facilitated through online networks, we are continuing to build sequencing capacity in Cameroon, Niger, and Nigeria, while also expanding our network to include other West African countries. Through this continued training and collaboration, we hope all participating groups will be able to sequence isolates locally from future cholera outbreaks and generate data from other high-priority pathogens. Finally, expanding sequencing knowledge, capacity, and collaboration beyond the original workshop participants will facilitate generation of data needed to control cholera outbreaks in the region.

## Acknowledgements

This project was made possible by the Nigeria Centre for Disease Control, who graciously hosted researchers at the National Reference Laboratory in Abuja, Nigeria for training and *V. cholerae* sequencing. We thank Aicha Omar, Zanguina Jibir, and Moussa Soussou Amadou for laboratory assistance and strain isolation, Alama Keita and the West Africa Cholera Platform for sharing incidence data, David Mohr for laboratory assistance and space, and Daryl Domman for providing *V. cholerae* alignments used as a background phylogenetic dataset. We also thank the original authors of the sequences used in this background dataset (references provided in *Figure 2—source data 3*). Funding for this project was provided by Bill and Melinda Gates Foundation OPP1195157.

## Additional information

### Funding

| Funder | Grant reference number | Author |
| --- | --- | --- |
| Bill and Melinda Gates Foundation | OPP1195157 | Andrew Azman<br>Justin Lessler |

The funders had no role in study design, data collection and interpretation, or the decision to submit the work for publication.

### Author contributions

Eme Ekeng, Conceptualization, Resources, Formal analysis, Investigation, Project administration, Writing - review and editing; Serges Tchatchouang, Blaise Akenji, Ibrahim Dan Dano, Conceptualization, Formal analysis, Investigation, Writing - original draft, Writing - review and editing; Bassira Boubacar Issaka, Ifeoluwa Akintayo, Christopher Chukwu, Sylvie Melingui, Michael Oladotun Popoola, Conceptualization, Formal analysis, Investigation, Writing - review and editing; Sani Ousmane, Supervision, Investigation, Writing - review and editing; Ariane Nzouankeu, Investigation, Writing - review and editing; Yap Boum, Francisco Luquero, Supervision, Writing - review and editing; Anthony Ahumibe, Dhamari Naidoo, Resources, Supervision, Project administration, Writing - review and editing; Andrew Azman, Justin Lessler, Conceptualization, Formal analysis, Supervision, Funding acquisition, Investigation, Visualization, Project administration, Writing - review and editing; Shirlee Wohl, Conceptualization, Software, Formal analysis, Validation, Investigation, Visualization, Writing - original draft, Project administration, Writing - review and editing

### Author ORCIDs

Andrew Azman (ID) http://orcid.org/0000-0001-8662-9077
Justin Lessler (ID) http://orcid.org/0000-0002-9741-8109
Shirlee Wohl (ID) https://orcid.org/0000-0002-0311-3348

### Decision letter and Author response

Decision letter https://doi.org/10.7554/eLife.65159.sa1
Author response https://doi.org/10.7554/eLife.65159.sa2

## Additional files

### Supplementary files

- Source code 1. Code used to generate figures.

- Supplementary file 1. Tabular results of antibiotic resistance testing and detection.

- Transparent reporting form

### Data availability

Raw data for all sequenced isolates is available under NCBI BioProject accession: PRJNA616029. Source data files have been provided for Figures 1 and 2, and R code used to make figures has been provided as Source Code File 1.

The following dataset was generated:

| Author(s) | Year | Dataset title | Dataset URL | Database and Identifier |
| --- | --- | --- | --- | --- |
| Ekeng E, Tchatchouang S, Akenji B, Issaka BB, Akintayo I, Chukwu C, Dano ID, Melingui S, Ousmane S, | 2020 | *Vibrio cholerae* whole genome sequencing | https://www.ncbi.nlm.nih.gov/bioproject/PRJNA616029 | NCBI BioProject, PRJNA616029 |

Popoola MO,
Nzouankeu A,
Boum Y, Luquero F,
Ahumibe A, Naidoo
D, Azman A, Lessler
J, Wohl S

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
