## [Decision Letter]

**Acceptance summary:**

The paper "Regional sequencing collaboration reveals persistence of the T12 *Vibrio cholerae* O1 lineage in West Africa" presents results from sequencing and analyzing 46 *Vibrio cholerae* whole genome sequence data. The paper presents findings from a region with little genomic surveillance, and as such these data are valuable. The study was conducted in the context of a regional training, and as such adds value as a potential model for regionally coordinated genomic surveillance efforts in areas where surveillance is limited.

**Decision letter after peer review:**

Thank you for submitting your article "Regional sequencing collaboration reveals persistence of the T12 *Vibrio cholerae* O1 lineage in West Africa" for consideration by *eLife*. Your article has been reviewed by 2 peer reviewers, and the evaluation has been overseen by a Reviewing Editor and a Senior Editor. The following individuals involved in review of your submission have agreed to reveal their identity: Surajit Basak (Reviewer #1); Taj Azerian (Reviewer #2).

Essential revisions:

The paper "Regional sequencing collaboration reveals persistence of the T12 *Vibrio cholerae* O1 lineage in West Africa" presents results from sequencing and analyzing 46 *Vibrio cholerae* whole genome sequence data. The paper presents findings from a region with little genomic surveillance, and as such these data are valuable. While the analysis doesn't provide much novelty in terms of understanding cholera transmission, the study was conducted in the context of a regional training, and as such adds value as a potential model for regionally coordinated genomic surveillance efforts in areas where surveillance is limited. However even though it seems that the authors aim to present this as a surveillance model, the current focus of the paper is on the somewhat limited inference made about transmission. We ask the authors to revise the paper, focusing on the regional genomic surveillance effort. In particular we would like the authors to provide more detail on:

1) Where did the isolates come from (e.g., cholera treatment centers, hospitals, or broader active surveillance)?

2) Do they conduct environmental sampling and could this be part of future efforts?

3) Who attended the training? Were they members from regional ministry of health labs, academic institutes etc?

4) Were the attendees laboratorians, bioinformaticians, clinicians etc?

5) Was there an effort to analyze the data, particularly the bioinformatics portion, locally or did the rely 100% on the collaborators at JHU? If the latter, this is probably not a good sustainable model for ongoing genomic epidemiology. If the prior, then were local or regional computing resources used

6) Are they continuing to sequence isolates after the training?

In addition we would like you to address the following points regarding the analyses:

7) A sentence about the cholera case definition would be helpful to clarify what reported suspected cases include. Is this based on symptoms, culture, molecular data?

8) Last paragraph of the introduction – There needs to be a little more preamble before introducing the lineage designations. How are they defined in the population structure of the 7th pandemic?

9) The authors should include some brief statistics on the distribution of regions represented in the 1280 comparison genomes even if these data can be abstracted from the supplemental material. These data would allow the reader to assess the representativeness of the sample. Particular focus should be given to the proportion of the sample from West Africa and the rest of the continent.

10) In the first paragraph of the discussion, it is stated that there is no evidence of introductions during the time periods in question. How was this inferred? Was it completely subjective or based on lineages and genetic distance? This should be clarified. Also, the authors address the limitation of sampling from the region. In particular, there is generally low representation from African countries. Further the T12 lineage is generally prevalent in the region. Therefore, if the determination of importations was made solely on lineages, then it will be difficult to resolve migration among the neighboring countries, as the authors note. Either there should be a more detailed analysis of strains from Africa (e.g., a separate phylogeny like the expanded clade in Figure 2) or clarification that importations are referring to migration from other regions where cholera is endemic e.g., SE Asia.

*Reviewer #1 (Recommendations for the authors):*

The authors may consider the following suggestions:

1. Is there any characteristics variation of mutational landscape of isolates from three different countries?

2. Does the mutational landscape change yearwise?

*Reviewer #2 (Recommendations for the authors):*

1) A sentence about the cholera case definition would be helpful to clarify what reported suspected cases include. Is this based on symptoms, culture, molecular data?

2) Last paragraph of the introduction – There needs to be a little more preamble before introducing the lineage designations. How are they defined in the population structure of the 7th pandemic?

3) The abstract is excessively sparse and could be expanded. It would be worthwhile to note that the analysis included 1280 other genomes as well as including the motivation for the analysis.

4) The authors should include some brief statistics on the distribution of regions represented in the 1280 comparison genomes even if these data can be abstracted from the supplemental material. These data would allow the reader to assess the representativeness of the sample. Particular focus should be given to the proportion of the sample from West Africa and the rest of the continent.

5) In the first paragraph of the discussion, it is stated that there is no evidence of introductions during the time periods in question. How was this inferred? Was it completely subjective or based on lineages and genetic distance? This should be clarified. Also, the authors address the limitation of sampling from the region. In particular, there is generally low representation from African countries. Further the T12 lineage is generally prevalent in the region. Therefore, if the determination of importations was made solely on lineages, then it will be difficult to resolve migration among the neighboring countries, as the authors note. Either there should be a more detailed analysis of strains from Africa (e.g., a separate phylogeny like the expanded clade in Figure 2) or clarification that importations are referring to migration from other regions where cholera is endemic e.g., SE Asia.

---

## [Author Response]

Essential revisions:The paper "Regional sequencing collaboration reveals persistence of the T12 *Vibrio cholerae* O1 lineage in West Africa" presents results from sequencing and analyzing 46 *Vibrio cholerae* whole genome sequence data. The paper presents findings from a region with little genomic surveillance, and as such these data are valuable. While the analysis doesn't provide much novelty in terms of understanding cholera transmission, the study was conducted in the context of a regional training, and as such adds value as a potential model for regionally coordinated genomic surveillance efforts in areas where surveillance is limited. However even though it seems that the authors aim to present this as a surveillance model, the current focus of the paper is on the somewhat limited inference made about transmission. We ask the authors to revise the paper, focusing on the regional genomic surveillance effort. In particular we would like the authors to provide more detail on:1) Where did the isolates come from (e.g., cholera treatment centers, hospitals, or broader active surveillance)?

We have added the following note to the methods describing where the isolates came from:

“In Cameroon, isolates came from cholera treatment centers from throughout the country. […] And in Nigeria, isolates came from cholera treatment centers in the Northeastern and Northwestern regions of the country.”

We have also added a mention of this in the newly added “Approach” section of the main text of the manuscript.

2) Do they conduct environmental sampling and could this be part of future efforts?

No environmental isolates were sequenced as part of this paper, as all included samples were collected as part of routine clinical surveillance. Sequencing from environmental samples could be a way to interrogate the hypothesis of an environmental reservoir. However, previous efforts have not been successful in identifying toxigenic O1 cholera in candidate reservoirs (e.g., in Lake Chad, as cited in the manuscript) so efforts here were focused on clinical isolates, as these have been shown to be more important for local and regional decision making. Building local capacity to sequence clinical isolates will likely prove more fruitful for understanding regional outbreaks in the short term, though sequencing from environmental samples will be an important future area of improvement for cholera research studies.

3) Who attended the training? Were they members from regional ministry of health labs, academic institutes etc?

Participants of the training included three individuals from Nigeria Centre for Disease Control National Reference Laboratory, one individual from Centre Pasteur du Cameroun, two individuals from the National Public Health Laboratory in Cameroon, two individuals from Centre de Recherche Médicale et Sanitaire in Niger, and one individual from the University of Ibadan in Nigeria. These participants brought with them a wealth of knowledge in regional cholera transmission, as well as microbiology techniques used in public health labs. We have restructured the manuscript to include an Approach section that more clearly discusses the participants and goals of the workshop, and we have added the following text to highlight the experience of the workshop participants:

“In October 2019, researchers from Cameroon, Niger, Nigeria, and the United States came together to discuss how whole genome sequencing could contribute to understanding of cholera spread in these countries. […] In all countries, reported cases were defined according to the World Health Organization suspected case definition (Global Task Force on Cholera Control Surveillance Working Group, 2017): individuals with acute watery diarrhoea and severe dehydration in areas where a cholera outbreak has not been declared, and individuals with acute watery diarrhoea in areas where an outbreak has been declared.”

4) Were the attendees laboratorians, bioinformaticians, clinicians etc?

As explained in the response above, most participants were microbiologists working in public health labs. All participants had technical laboratory skills, and a few had prior bioinformatics experience. We hope that the added Approach section of the manuscript provides additional details as to the goals and participants in the workshop experience.

5) Was there an effort to analyze the data, particularly the bioinformatics portion, locally or did the rely 100% on the collaborators at JHU? If the latter, this is probably not a good sustainable model for ongoing genomic epidemiology. If the prior, then were local or regional computing resources used

Absolutely. The workshop provided a basic introduction to sequencing processes (basecalling, demultiplexing, assembly) as well a tutorial on using the Linux command line to run bioinformatics processes. During the workshop, participants worked on one of two provided sequencing laptops (one of which remains at the National Reference Laboratory at Nigeria Centre for Disease Control) or own their own laptop booted from an external hard drive with all software installed.

It is very difficult to become a bioinformatics expert overnight, so the workshop focused on the key concepts needed to be conversant in whole genome sequencing and bioinformatics, in the hopes that this would seed future collaborations between the workshop participants and with other groups. Indeed, Nigeria Centre for Disease Control is currently involved in several pathogen sequencing projects (notably including ongoing SARS-CoV-2 sequencing projects), several using the laptop and MinION (and bioinformatics protocols) provided during the workshop. The workshop also focused on using publicly available phylogenetic tools and on interpretation of phylogenetic trees, and we spent the last day of the workshop discussing the conclusions from a phylogenetic tree that included the sequences generated during the training.

Additional follow up phylogenetic analysis for this manuscript did rely on collaborators at JHU, but all participating groups have been invited to an ongoing follow-up workshop, which will again teach key bioinformatics skills and analysis techniques. All of these analyses are to be conducted on laptop computers provided to participating groups (or acquired independently, with advice and setup support from JHU collaborators), with the goal of working towards a more sustainable model for ongoing genomic epidemiology that does not rely on JHU.

Finally, all antibiotic resistance analyses presented in the manuscript were conducted entirely independently from the JHU authors, using the FASTA files generated during the workshop and local expertise.

As shown in the expert copied above, we have aimed to revise the manuscript to highlight the emphasis on bioinformatics training and collaborative analysis during the workshop period. We have also added additional text to the discussion highlighting some of the points above:

“As a result of the workshop, scientists from Cameroon, Niger, and Nigeria have all been trained in V. cholerae sequencing with the Oxford Nanopore MinION platform, and scientists at Nigeria Centre for Disease Control National Reference Laboratory have used this expertise to continue sequencing *V. cholerae* and other pathogens. […] Finally, expanding sequencing knowledge, capacity, and collaboration beyond the original workshop participants will facilitate generation of data needed to control cholera outbreaks in the region.”

6) Are they continuing to sequence isolates after the training?

As mentioned briefly above, Nigeria Centre for Disease control has continued to sequence isolates after the training. They briefly attempted cholera sequencing but have mostly transitioned to SARS-CoV-2 sequencing in light of the ongoing pandemic. Individuals from Centre Pasteur du Cameroun, the Cameroon National Public Health Lab, and Centre de Recherche Médicale et Sanitaire in Niger recently received sequencing equipment (including a laptop for bioinformatic analysis) and will begin sequencing SARS-CoV-2 as part of an ongoing follow-up workshop. The hope is that additional training in pathogen sequencing will contribute to cholera surveillance after the imminent needs of the COVID-19 pandemic have subsided. As shown in the previous response, we have updated the manuscript to mention ongoing training and opportunities to build sequencing capacity (despite the brief pivot away from cholera in light of the ongoing COVID-19 pandemic).

In addition we would like you to address the following points regarding the analyses:7) A sentence about the cholera case definition would be helpful to clarify what reported suspected cases include. Is this based on symptoms, culture, molecular data?

We have added a sentence on how cholera cases were defined to the Approach section and the Figure 1 caption. Specifically, we specify that case counts are clinically based, using the WHO definition of suspected cases:

“In all countries, reported cases were defined according to the World Health Organization suspected case definition (Global Task Force on Cholera Control Surveillance Working Group, 2017): individuals with acute watery diarrhoea and severe dehydration in areas where a cholera outbreak has not been declared, and individuals with acute watery diarrhoea in areas where an outbreak has been declared.”

8) Last paragraph of the introduction – There needs to be a little more preamble before introducing the lineage designations. How are they defined in the population structure of the 7th pandemic?

This is an excellent point, and we realize we did not provide enough information about the existing cholera lineages in our first submission. We have restructured the introduction of the manuscript entirely, now including an Approach section, but have made sure to include this background information in the introduction:

“One way to distinguish between transmission scenarios is to compare the specific *V. cholerae* lineages in circulation. […] In this manuscript, we aim to use additional sequence data from recent cholera outbreaks to improve our understanding of more detailed *V. cholerae* transmission patterns within West Africa, and to understand if regionally coordinated surveillance and response may improve containment outcomes.”

9) The authors should include some brief statistics on the distribution of regions represented in the 1280 comparison genomes even if these data can be abstracted from the supplemental material. These data would allow the reader to assess the representativeness of the sample. Particular focus should be given to the proportion of the sample from West Africa and the rest of the continent.

The majority of the background samples included in this study come from Genomic history of the seventh pandemic of cholera in Africa published by Weill et al. in 2017. The Weill et al. publication includes a breakdown of the 1070 sequences included in that paper by introduction (Figure 3) and by continent and year (Figure S1). They also include a plot of the reported cholera cases in Africa by year in Figure S2, for purposes of comparison.

To facilitate understanding of the background dataset in this manuscript, we have added Figure 2—figure supplement 2, which breaks down the included sequences by decade and continent, and further enumerates sequences from the regions and countries relevant to this manuscript. We have also added Figure 2—figure supplements 3 and 4 in order to compare the sequences included in this study to cases reported to the WHO. In each, we report the distribution of cases across regions for the relevant time period, and plot this alongside the distribution of sequences across regions for the relevant time period. As shown in these figures, the African continent is well represented by sequence data across all relevant time periods. Additionally, Western and Central Africa – the regions most relevant to this study – are consistently well represented by the included sequence data (though the total number of sequences are of course limited in number).

To the Results section, we have added the following description:

“We aligned these genomes to 1280 previously published *V. cholerae* O1 whole genome sequences (Bwire et al., 2018; Irenge et al., 2020; Weill et al., 2019, 2017), which provide a broad overview of global seventh pandemic O1 cholera diversity. The majority of these sequences are from Africa and South Asia, and reported cases in West and Central Africa (World Health Organization, 2019) are well represented by the sequence data (Figure 2—figure supplements 2–4).”

10) In the first paragraph of the discussion, it is stated that there is no evidence of introductions during the time periods in question. How was this inferred? Was it completely subjective or based on lineages and genetic distance? This should be clarified. Also, the authors address the limitation of sampling from the region. In particular, there is generally low representation from African countries. Further the T12 lineage is generally prevalent in the region. Therefore, if the determination of importations was made solely on lineages, then it will be difficult to resolve migration among the neighboring countries, as the authors note. Either there should be a more detailed analysis of strains from Africa (e.g., a separate phylogeny like the expanded clade in Figure 2) or clarification that importations are referring to migration from other regions where cholera is endemic e.g., SE Asia.

We have rephrased the first sentence of the discussion to clarify that lack of evidence for introductions is based on the fact that all newly sequenced isolates from the region fall clearly within the T12 lineage, which was the lineage last reported in the region. We have also specified that we mean introduction from outside of West Africa, since the reviewer correctly notes that we are not able to resolve transmission within the region:

“We found no evidence for a new introduction of *V. cholerae* O1 from outside of West Africa between 2014 – the date of the last published isolate from these countries – and 2018. […] In concordance with the epidemiologic data, the high similarity between sequences (including shared mutations in known AMR genes, see Supplementary File 1A and 1B) from the three countries suggests that their outbreaks are linked.”

Reviewer #1 (Recommendations for the authors):The authors may consider the following suggestions:1. Is there any characteristics variation of mutational landscape of isolates from three different countries?

As shown in the phylogeny, 2018–2019 sequences from Cameroon, Niger, and Nigeria form a distinct cluster. This cluster shares 4 mutations not seen in any other T12 sequences. However, the fact that these mutations are not shared with previously published sequences from the same countries suggests that these mutations are not characteristic to the countries, but rather the result of evolution of the bacterium over time (as noted in the Discussion section). Additionally, the sequences from the three countries are interspersed within the 2018–2019 cluster, indicating there are no mutations characteristic of one country and not seen in the others (confirmed by analysis of individual variants in these sequences). These observations support our conclusion that outbreaks in these three countries are likely linked.

2. Does the mutational landscape change yearwise?

While this is an important question, we do not have enough samples from each year to provide a definitive answer. We do note in the discussion, however, that samples from 2018–2019 form a distinct clade on the phylogeny (distinct from 2014 and 2016), which suggests evolution of the mutational landscape over this time period.

Reviewer #2 (Recommendations for the authors):The abstract is excessively sparse and could be expanded. It would be worthwhile to note that the analysis included 1280 other genomes as well as including the motivation for the analysis.

The reviewer is correct. We have revised the abstract accordingly. It now reads:

“Despite recent insights into cholera transmission patterns in Africa, regional and local dynamics in West Africa – where cholera outbreaks occur every few years – are still poorly understood. […] The results from this regional sequencing effort provide a model for future regionally coordinated surveillance efforts.”